# Antimicrobial Resistance of *Salmonella* Strains Isolated from Human, Wild Boar, and Environmental Samples in 2018–2020 in the Northwest of Italy

**DOI:** 10.3390/pathogens11121446

**Published:** 2022-11-30

**Authors:** Valeria Listorti, Aitor Garcia-Vozmediano, Monica Pitti, Cristiana Maurella, Daniela Adriano, Carlo Ercolini, Monica Dellepiane, Lisa Guardone, Elisabetta Razzuoli

**Affiliations:** 1Section of Genoa, Istituto Zooprofilattico Sperimentale del Piemonte, Liguria e Valle d’Aosta, Piazza Borgo Pila 39/24, 16129 Genoa, Italy; 2Section of Turin, Istituto Zooprofilattico Sperimentale del Piemonte, Liguria e Valle d’Aosta, Via Bologna 148, 10154 Turin, Italy; 3Section of Savona, Istituto Zooprofilattico Sperimentale del Piemonte, Liguria e Valle d’Aosta, Via Martiri 6, 17056 Savona, Italy; 4Section of La Spezia, Istituto Zooprofilattico Sperimentale del Piemonte, Liguria e Valle d’Aosta, Via degli Stagnoni 96, 19100 La Spezia, Italy

**Keywords:** food-borne pathogens, *Salmonella*, surveillance, antimicrobial resistance, One Health

## Abstract

Antimicrobial resistance is one of the most challenging public health problems worldwide, and integrated surveillance is a key aspect in a One Health control strategy. Additionally, *Salmonella* is the second most common zoonosis in Europe. We aimed to investigate the circulation of *Salmonella* strains and their related antimicrobial resistance in human, environmental, and wild boar samples from the northwest of Italy, from 2018 to 2020, to obtain a more comprehensive epidemiological picture. *Salmonella* Typhimurium 1,4,[5],12:i:-, *S.* Veneziana and *S.* Newport were the most common serotypes occurring in humans, the environment, and wild boar, respectively. Antimicrobial resistance was rather common in *Salmonella* isolates, with those from human displaying the highest degree of resistance against sulfadiazine–sulfamerazine–sulfamethazine (>90% of resistance). Moreover, resistance against azithromycin were exclusively observed in environmental samples, while only 7.7% (95% CI = 1.6–20.8) of wild boar isolates experienced resistance against trimethoprim–sulfamethoxazole. Multidrug resistance concurrently involved up to seven antimicrobial classes in human isolates, including third-generation cephalosporins and fluoroquinolones. *Salmonella* Typhimurium in humans and serotypes Goldcoast and Rissen from environmental sources showed the highest levels of resistance. This study shows diverse antimicrobial resistance patterns in *Salmonella* strains isolated from different sources and gives a broad picture of antimicrobial resistance spread in wild animals, humans, and the environment.

## 1. Introduction

Salmonellosis is the second most common zoonosis reported in Europe [1]. The genus *Salmonella* comprises enteric bacteria hosted by humans and numerous domestic and wild animals [2]. Some *Salmonella* serovars, such as *S.* Typhi and *S.* Parathypi A and C, are host-adapted and responsible for typhoid fever in humans; they, in turn, act jointly, with primates as their main reservoirs [3,4]. Other serovars have a generalist behaviour, infecting a broad range of hosts, and are mainly responsible for non-typhoidal diseases in humans. Among these, *S.* Enteritidis, *S.* Typhimurium, *S.* Typhimurium 1,4,[5],12:i:-, *S.* Infantis and *S.* Derby have been recently reported to be the most common serotypes involved in human infections [1]. *Salmonella* is excreted via the faecal route and is able to disseminate in the environment and enter in the food chain. Salmonellosis in humans is generally contracted through the consumption of contaminated food of animal origin, but other types of foods, such as green vegetables irrigated with contaminated water, might also serve as a vehicle of infection [5]. 

Antimicrobial resistance (AMR) in foodborne pathogens are of great concern for public health safety [6], especially for the emergence of multidrug-resistant (MDR) strains, i.e., bacteria showing resistance to at least three or more antimicrobial classes [7]. Many factors may favour their development and spread, including the inappropriate use of antimicrobials in human medicine, self-medication or the early suspension of prescribed therapies, as well as their misuse in food-producing animals, aquaculture and agriculture [8]. Genes conferring resistance can be transferred among bacteria by different mechanisms [9], increasing the abundance of resistant pathogenic microorganisms that, in turn, potentially hamper the effectiveness of antimicrobial treatments. Understanding the AMR dynamics between animals and humans sometimes becomes complicated, especially when they share the same environment. The latter, alongside wild fauna, can play an important role as reservoirs of AMR, contributing to the spread of resistant bacteria to food-producing animals and humans [10]. Among wildlife species, wild boar can serve as a good indicator for the environmental spread and transmission of resistant *Salmonella* strains [11]. In fact, this species is increasingly expanding to more anthropized areas, leading to direct and indirect contacts with humans and their domestic animals that may also increase the chance of disease transmission [12]. Wild boar is the most widespread wild ungulate species in Liguria, northwest Italy, sometimes concentrating in high densities even near peri-urban and urban areas.

Integrated surveillance of AMR in pathogenic bacteria spreading in humans, animals and the environment, is one of the top priorities for the European Union/European Economic Area (EU/EEA) [13]. In 2003, the European Parliament and the Council issued Directive 2003/99/EC to ensure that zoonoses, zoonotic agents, and related AMR were properly monitored [14]. In Italy, the surveillance of *Salmonella* in humans and in the environment is coordinated by Enter-Net Italia (*Istituto Superiore di Sanità*, Rome), while its surveillance in food products and animals is organized by the Italian Reference Centre for Salmonellosis (Enter-Vet, *Istituto Zooprofilattico Sperimentale delle Venezie*, Padova). In Liguria, *Salmonella* surveillance is carried out by the Reference Centre for *Salmonella* Typing (CeRTiS, *Istituto Zooprofilattico Sperimentale del Piemonte, Liguria e Valle d’Aosta*, Turin), which collects *Salmonella* strains isolated from human, animal, and environmental samples according to national surveillance and monitoring programs. In the view of a One Health approach, this study aimed to investigate the serotypes and related AMR profiles of *Salmonella* spp. strains circulating in humans, the environment and wild boar in Liguria, northwest of Italy, between 2018 and 2020.

## 2. Results

### 2.1. Salmonella Serotypes and Subtypes Identification

A total of 517 *Salmonella* specimens were collected in 2018–2020 in Liguria, including 264 strains isolated from human samples, 193 from the environment, and 60 from wild boar livers. Four different subspecies of *Salmonella enterica* occurred among positive samples, though *S. enterica* subsp. enterica was the main subtype detected (94.2%; 95% CI = 91.8–96.1). Other subspecies detected included *S. enterica* subsp. diarizonae (4.1%; 95% CI = 2.5–6.1), *S. enterica* subsp. salamae (1.2%; 95% CI = 0.4–2.5), and *S. enterica* subsp. houtenae (0.6%; 95% CI = 0.1–1.7). *Salmonella* isolates generally showed great diversity in serotypes, especially in human infections (Pearson’s Chi squared test, *p* < 0.001), in which 34 different serotypes were identified. *Salmonella* Typhimurium 1,4,[5],12:i:- was by far the most prevalent strain involved in symptomatic human patients (Figure 1), followed by serotypes Typhimurium, Napoli, Enteritidis, Rissen, and Infantis, which accounted for 30.7% (95% CI = 25.1–36.6) of infections. Minor serotypes totalized 20.3% (95% CI = 15.6–25.7) of human infections, ranging between 0.4% and 1.9% prevalence (Table 1). 

A higher number of inconclusive results in *Salmonella* typing were obtained from environmental (*n* = 35) and wild boar (*n* = 10) isolates. As opposed to human *Salmonella*, environmental and wild boar isolates experienced lower levels of diversity. We recovered a total of 22 different serotypes from the environment and 13 serotypes occurring in wild boar infections. Serotypes Veneziana, Napoli, and Stourbridge prevailed in tested-positive samples from the environment, while serotypes Newport and Livingstone dominated, along with the subspecies diarizonae, among wild boar infections (Figure 1).

Twenty-two out of 35 *Salmonella enterica* subspecies and serotypes were shared between different sources. In particular, serotypes Typhimurium 1,4,[5],12:i:-, Napoli, Newport, Stourbridge and *S. enterica* subsp. diarizonae simultaneously occurred in humans, wild boar infections and in environmental samples. However, this phenomenon more often occurred among human and environmental isolates (n= 12; Table 1). By contrast, a low proportion of *Salmonella* from wild boar occurred in both humans (n= 3) and environmental isolates (*n* = 2).

### 2.2. Antimicrobial Resistances in Salmonella Isolates

Out of 517 isolates collected, we examined only 415 isolates for which complete data on AMR were available. In particular, we excluded 22.3% (*n* = 102) of isolates that were analysed during the last half of 2020, following a different routine test method, thus preventing any comparisons with the rest of the *Salmonella* isolates in the study. The excluded specimens included 47 *Salmonella* collected from the environment, 34 from human patients, and 21 from wild boar. 

Susceptibility testing showed that nearly all isolates appeared susceptible to at least one antibiotic (*n* = 411). Intermediate susceptibility was jointly observed in 43.1% (95% CI = 38.3–48.1) of human and environmental isolates but not in wild boar strains. Moreover, intermediate results mainly predominated in humans (*n* = 114/230; *p* < 0.05) compared with environmental isolates (*n* = 65/146). Antimicrobial resistance was fairly common among *Salmonella* isolates, with an overall resistance prevalence of 62.9% (*n* = 261; 95% CI = 58.0–67.6). However, we observed clear differences on AMR levels according to the isolate origin (*p* < 0.001). In fact, human isolates typically recorded the highest levels of AMR (80.4%; 95% CI = 74.7–85.4) compared with those recovered from the environment (50.0%; 95% CI = 41.6–58.4) and from wild boar (7.7%; 95% CI = 1.6–20.8). 

Resistance against single-drug molecules occurred in 33.4% (95% CI = 25.8–37.4) of the *Salmonella* isolates, involving 9 out of 16 antimicrobials tested. The combination of sulfadiazine–sulfamerazine–sulfamethazine (SSS) generally displayed the highest degree of resistance (Figure 2), with an average prevalence of 86.3% (95% CI = 80.5–91.0). This was plainly evident in most of the human isolates (Table 2; *p* < 0.001) and less so for environmental isolates. Moreover, the exposure to trimethoprim–sulfamethoxazole (SXT), tetracycline (TET), and streptomycin (STR) yielded similar results (Figure 2); nonetheless, AMR levels against TET and STR differed between human and environmental isolates (Table 2; *p* < 0.001). In the case of wild boar isolates, only AMR levels against SXT were detected (Table 2). Resistance against ampicillin (AMP) and tigecycline (TGC) were likewise relevant, especially in humans (*p* < 0.001). Despite the reduced number of isolates tested against azithromycin (AZI), resistant patterns were only observed among environmental strains (Table 2). Few human isolates were proven to be resistant to the effect of chloramphenicol (CHL), while environmental isolates were totally vulnerable to its activity. Low AMR levels occurred against amoxicillin + clavulanic acid (AMC) and gentamicin (GEN). Conversely, meropenem (MER) was the unique antibiotic for which no resistant isolates were detected (Table 2). Isolates exposed to the tested cephalosporins displayed high susceptibility levels (Figure 2), with AMR prevalence ranging between 0.4% and 5.1% in human isolates. Amongst quinolones, nalidixic acid (NAL) showed moderate levels of AMR (11.2%; 95% CI = 6.5–17.5) in human isolates, while lower levels predominated for ciprofloxacin (CIP).

Resistance against the combination of two types of antibiotics occurred in 13.8% (95% CI = 10.5–17.7) of *Salmonella* isolates, yielding 15 different combinations. Among these, only three concurrently occurred among human and environmental isolates, including SSS–TET, SSS–STR, and TET–TGC (Appendix A). 

### 2.3. Multi Drug Resistance in Salmonella Isolates 

Multi-drug resistant patterns (≥three molecules of different classes) broadly spread among a great proportion of *Salmonella* isolates, though they were more frequently noted among those collected from humans (58.8%; 95% CI = 50.9–65.6; *p* < 0.001). The phenomenon also occurred among environmental strains (13.7%; 95% CI = 6.8–23.8), while no MDR isolates were detected from wild boar. We obtained a total of 50 MDR profiles, of which only one was shared between *S.* Typhimurium 1,4,[5],12:i:- recovered from both humans and the environment and involved four different antimicrobial classes (AMP, STR, SSS, and TET). Human strains concurrently recorded MDR to up to seven types of antibiotics, while MDR levels in environmental isolates reached up to five antibiotics. 

Maximum levels of MDR in humans occurred in one isolate of *S.* Typhimurium (AMP-TAZ-CIP-CHL-STR-SSS-TET-NAL). Notwithstanding, the highest prevalence of MDR were recorded in *S.* Typhimurium 1,4,[5],12:i:- isolated from humans (63.9%; 95% CI = 54.1–72.9). Among environmental isolates, serotypes Goldcoast and Rissen displayed the maximum MDR levels with the following combinations: AZI-CIP-TET-TGC-STX and AZI-GEN-TET-TGC-SXT, respectively. Gentamycin was the antibiotic that was most engaged in MDR combinations of three molecules, especially in environmental isolates. By contrast, SSS, STR, TET, and AMP most prevailed in the MDR observed in human isolates (59.1%; 95% CI = 36.4–79.3). The contribution of TGC, AZI, and CHL to MDR was quite low, often corresponding to single cases (Appendix A). Cephalosporins and quinolones were also involved in MDR, contributing to 21.2% (95% CI = 14.2–29.7) of the profiles. These antibiotics prevailed among MDR over four antimicrobials (*n* = 11/25) to up to the maximum levels reported above. Antimicrobial resistance profiles to which cephalosporine and quinolones contributed are summarized in Table 3. The main serotypes involved in AMR against cephalosporines and quinolones included: S. Typhimurium 1,4,[5],12:i:-, which often displayed resistance against the three cephalosporines in the study; the serotypes Infantis and Rissen, whose AMR profiles involved the quinolones NAL and CIP; and serotypes Typhimurium and Othmarschen, which displayed resistance to both cephalosporines and quinolones (Table 3). 

## 3. Discussion

This study highlights the great diversity of *Salmonella* spp. to which the human population is exposed in Liguria, northwest Italy, as well as the main serotypes circulating in its natural environment, including the wild boar population. *Salmonella* Typhimurium 1,4,[5],12:i was the predominant serotype identified in human infections, which is in contrast to the recent data reported about zoonoses published at the European level by the European Food Safety Authority (EFSA) and the European Centre for Disease Prevention and Control (ECDC) [1]. In 2020, *S.* Typhimurium 1,4,[5],12:i ranked in the third position among the most prevalent serotypes involved in human infections in Europe, with around 11% prevalence [1]. In Italy, this serotype has been frequently reported to be the most common cause of salmonellosis in humans and the main serotype isolated from animals and meat food products, especially in those from the swine production chain [15,16]. Infection by *S*. Typhimurium 1,4,[5],12:i has been recently recorded to exceed up to four times the infection by *S*. Typhimurium in humans, animals and food products [16]. This trend also emerged among the human infections occurring in our study area, where *S.* Typhimurium ranked in second place, with prevalence comparable to those reported at the European level [1]. Differences in the predominance of *Salmonella* serotypes in human infections may be explained by differences in the geographical context. For instance, *S*. Typhimurium seems to be the most prevalent serotype involved in human infections from southern Italy, prevailing over *S*. Typhimurium 1,4,[5],12:i: and *S.* Enteritidis [17]. 

*Salmonella* Napoli was the third serotype recorded in human infections during the study period and the second most common serotype isolated from the environment. The occurrence of this serotype is increasing in Italy but is uncommonly reported in Europe [18,19,20]. We isolated *S*. Napoli also from wild boar, supporting previous findings reported from the same area that highlighted their potential role as spreaders of this serotype in the environment [21]. 

*Salmonella* Enteritidis is the most common serotype involved in human infections in Europe [1]; however, we reported a lower prevalence, being the fourth most common serotype identified in human infections. In the veterinary field, a notable reduction in the presence of this serotype has been generally observed in recent years in Italy. The active Italian control programs implemented for the eradication of *Salmonella* in poultry farms may justify the minor contribution of *S.* Enteritidis to human infections [15,22]. Similarly, other European countries such as Greece, endowed with comparable eradication programs against *Salmonella* in poultry, have observed the decreasing trend of *S*. Enteritidis in humans, reducing its prevalence by half in the last 20 years [23]. Notwithstanding, in this country, *S.* Enteritidis remains the main serotype isolated in human samples. 

*Salmonella* Infantis and *S.* Derby are among the top five serotypes involved in human infections in Europe. The epidemiological situation observed for these serotypes in the Ligurian population is in agreement with the European average [1] and with that experienced in other Italian regions [17]. An increasing trend of infection by *S.* Infantis has been recently described in poultry meat [24,25], and it has also been associated with human outbreaks in Italy [26,27], while human infections by *S.* Derby are mostly related to the consumption of pork and poultry products [28]. *S.* Typhi, responsible for typhoid fever in humans, was isolated in a very low percentage of the human cases in our study. Despite typhoid fever being rare in Italy, mainly occurring in people travelling from endemic countries, important levels of AMR were observed in isolated strains [29]. 

The lower diversity in serotypes observed in environmental strains may be justified by the lower sampling effort employed. However, environmental factors, such as precipitation and water temperature, strongly influence the abundance and diversity of *Salmonella* spp. in surface waters [30]. *Salmonella* Veneziana was the most common serotype isolated from surface waters, followed by *S.* Napoli and *S.* Stourbridge. We also reported one isolate of *S.* Veneziana from human samples. This serotype has been previously reported to cause human infections [31] and has also been isolated from wild animals [15,32,33,34]. We also ascertained the presence of both *S*. Typhimurium and *S*. Typhimurium 1,4,[5],12:i in water surface samples although in low prevalence. By contrast, a previous study investigating the diversity of *Salmonella* strains in environmental water in France reported *S.* Typhimurium as the most widespread serotype in river water, marine, and freshwater sediment, probably related to the wastewater discharge from animal-rearing activities [35]. Our findings about the low presence of *S.* Typhimurium and its monophasic variant in the sampled surface waters could be an indicator of water contamination from not only humans but from other sources, such as animal faeces. For instance, our research group has previously reported *S.* Veneziana, *S.* Napoli, and *S.* Stourbridge in wild boar samples in the same area of study [21]. The *S.* Stourbridge has been proven to cause disease in humans [36] and its additional discovery in wild boar samples reaffirms the possible role of this animal species in the transmission of this serotype. *S.* Newport predominated among the serotypes identified from wild boar. Although this latter serotype was not relatively common among the human cases investigated, its pathogenicity has been proven in other countries by being the main one responsible for several human outbreaks [37,38]. 

We generally uncovered high AMR levels in our sample of *Salmonella* isolates. The highest prevalence of AMR was especially observed against sulfamides, ampicillin, tetracycline, and streptomycin, as previously described [39]. The levels of AMR reported for these antibiotics are even higher than those reported for the general Italian context [40], excepting for streptomycin, for which data are not available. Our findings are consistent with the results obtained from a recent Italian survey, in which resistance against sulfamides, ampicillin, and tetracycline were the most common type of AMR observed in human *Salmonella* isolates [17]. An increase in AMR against ampicillin, chloramphenicol, and trimethoprim–sulfamethoxazole—i.e., antibiotics previously considered of first line for the treatment of *Salmonella* infections—has been globally observed [41]. 

We also observed a high level of resistance against tigecycline among human isolates. This antibiotic is considered an optimal choice for the treatment of MDR *Salmonella* infections; however, its efficacy may be compromised due to the phenomenon of heteroresistance, which has been already observed for *S*. Typhimurium [42]. Despite the low resistance levels usually reported for human *Salmonella* strains against tigecycline [17,39,40], in some contexts these levels of resistance are higher and spread in some livestock productions, such as poultry [43,44]. 

Fluoroquinolones are recommended drugs to treat invasive salmonellosis or fragile patients at risk of developing it, but AMR in typhoid and non-typhoidal Salmonellae against this antibiotic is increasingly reported [45,46,47]. In 2017, the World Health Organization (WHO) listed fluroquinolones-resistant Salmonellae among the high-priority-bacteria group for which it is necessary to support the research and development of effective drugs [48]. The AMR observed against nalidixic acid in our human strains are in line with reports from other Italian and European regions, while the reported AMR prevalence against ciprofloxacin is lower [17,39,40]. 

Third-generation cephalosporins are defined by WHO within the category of "highest priority critically important antimicrobials". Variable degrees of AMR against this kind of antibiotics have been observed in typhoid and non-typhoidal *Salmonella* strains involved in human infections worldwide [49,50,51]. The origin of AMR against third-generation cephalosporins mainly resides in extended-spectrum β-lactamase genes [52]. In Europe, an increase of this kind of AMR has been recently reported, especially in *Salmonella* strains isolated from humans [52]. Notwithstanding, the average prevalence of AMR against third-generation cephalosporins (ceftazidime and cefotaxime) remains at least relatively low (0.8%). For the Italian context, the AMR prevalence against ceftazidime and cefotaxime are similar, averaging around 2% for both antibiotics [40]. We have recorded higher AMR levels against ceftazidime, while a lower prevalence was observed for cefotaxime. However, contrasting results have been reported in other regions. For instance, in the Netherlands, there have been observed higher levels of resistance against cefotaxime and lower AMR levels against ceftazidime, recording a maximum prevalence of 2.1% [39]. In the south of Italy, resistance levels were analysed, targeting specific *Salmonella* serotypes: *S*. Derby, *S*. Infantis, and *S.* Typhimurium, and recorded the highest levels of AMR against ceftazidime and cefotaxime, ranging from 19.0% to up to 40.0% of prevalence, however, the greatest levels of AMR were generally recorded when these serotypes were exposed to cefotaxime [17]. We observed resistance against cephalosporins of the third generation in 12 different strains, but they most frequently occurred in *S.* Typhimurium 1,4,[5],12:i:-. 

An important finding of this study is the absence of resistance against carbapenem in all of the *Salmonella* strains isolated; they are considered critical antibiotics of last resort. Carbapenem-resistant *Salmonella* have been previously isolated from animals, animal-food products, and humans [53,54,55]. In Europe, only a few official reports related to carbapenem resistance have been hitherto published, recording a very low level of AMR in the human infections from Spain (0.1%) in 2019 and from Denmark (0.4%) and Belgium (0.1%) in 2020 [40]. 

We also observed noteworthy AMR levels against azithromycin that exclusively involved almost all environmental strains isolated in 2020. Our results, however, cannot be directly compared with other studies, since our sample size was very low. In Europe, azithromycin resistance has been reported in human isolates from 2020, with an average prevalence of 0.8%; in Italy, this prevalence reached levels up to 0.3% during the same period. Even so, records from southern Italy have reported 36.3% of AMR prevalence against azithromycin in *Salmonella* strains isolated from humans and a lower percentage in samples from food of animal origin (7.3%) between 2017 and 2021 [17]. In 2020, azithromycin was overused during the COVID-19 pandemic [56,57,58], so it would be interesting to monitor the trend of the AMR against this antibiotic to evaluate its impact in the near future. 

We observed that the phenomenon of MDR more often occurred among human isolates; however, simultaneous resistance against ampicillin, streptomycin, tetracycline, and sulfamides (AMP-STR-SSS-TET) was experienced by *S.* Typhimurium 1,4,[5],12:i: isolated from both humans and the environment. This co-resistance is increasing worldwide, and it has also been recently reported in Italy, especially in *Salmonella* strains collected from human and animal samples [59,60].

The spread of *Salmonella* strains, concurrently resistant to third-generation cephalosporines and fluoroquinolones, is an important therapeutic issue in many parts of the world [61,62]. Unfortunately, we also observed co-resistance to ciprofloxacin and ceftazidime in two isolates of *S.* Typhimurium and in a single *S.* Othmarschen of human origin. 

Although moderate and high levels of AMR were observed against the combination trimethoprim + sulfamethoxazole and sulfamides, the AMR prevalence observed in environmental strains was generally lower compared with that detected in human samples. High AMR prevalence against sulfamides and streptomycin have been reported in urbanized areas [63], as has the presence of environmental strains resistant against ciprofloxacin and cefotaxime [48,59]. In our environmental strains, no evidence for resistance against third-generation cephalosporines was observed, while the AMR levels against fluoroquinolones were very low. The role of the environment in the spread of AMR is known and has been recently reviewed [64]. Most of the AMR and MDR strains detected from our environmental isolates are probably related to water contamination, for example, as a consequence of wastewater discharge from anthropized areas and/or agricultural activities, but also by the entry of *Salmonella* strains from wildlife that may host AMR strains of variable degree. 

Many actions have been taken by the European Union and the Italian government to face the AMR phenomenon. The reduction in the use of antimicrobials in animals has been one of the main targets of these initiatives, in particular in livestock [65]. In our geographic context, livestock activity is very limited; hence, its contribution to the spread of AMR pathogens can be considered low. This situation may explain the lower presence of AMR in the environmental samples and also the high susceptibility observed in *Salmonella* isolates from wild boar. Conversely, the role of wild boars as reservoir and spreaders of MDR bacteria had been already observed in the same study area [15,66] but also in other parts of Italy [67,68] and Europe [69]. Herein, we report lower levels of AMR in wild boar compared to previous studies conducted in the same study area [21]. These contrasting results may be explained by the lower sample size analysed in the present study, making it difficult to draw a general picture about AMR in this animal population. More in-depth studies should be carried out in order to better characterize AMR strains circulating in wild boar populations and to assess the role of the environment in the release of resistant bacteria. 

## 4. Materials and Methods

### 4.1. Strains

A total of 517 *Salmonella* spp. strains isolated from different sources, including humans, animals and the environment in 2018–2020, in Liguria, were used in this study. From these, 264 *Salmonella* strains were isolated from human patients showing clinical signs referring to salmonellosis and delivered to the *Istituto Zooprofilattico Sperimentale of Piemonte, Liguria and Valle D’Aosta* by different hospitals in Liguria. These strains originated from different kind of samples, including faeces (*n* = 240), urine (*n* = 15), and blood (*n* = 9). A total of 193 environmental strains, originating from samples of surface waters, were conferred by the Regional Agency for Environmental Protection of Liguria. Regarding wild boar, sixty *Salmonella* strains were isolated in our laboratory from liver samples collected in hunted animals from 2018 to 2020, within the frame of annual monitoring plans on wildlife health and the food safety of derived products. The isolation of *Salmonella* was performed from 25 g of each liver using a previously described procedure [21] according to ISO 6579:2002/COR. 1, 2004 (Microbiology of food and animal feeding stuffs: horizontal method for the detection of *Salmonella* spp.). Serotype identification was carried out using the standard agglutination method, according to ISO/TR 6579-3, 2014 (Microbiology of the food chain—Horizontal method for the detection, enumeration and serotyping of Salmonella-Part 3: Guidelines for serotyping of *Salmonella* spp.). 

### 4.2. Antimicrobial Resistance Analyses

Antimicrobial resistance were investigated by the Kirby–Bauer disk diffusion test using Mueller–Hinton agar plates (Microbiol, Uta, Italy). Results were interpreted and categorized in three classes (susceptible, intermediate, and resistant) according to Clinical and Laboratory Standard Institute (CLSI) guidelines [70], except for tigecycline, for which FDA-identified interpretative criteria were followed [71]. Different panels of antimicrobials were used for susceptibility testing depending on the source of isolates; sometimes the isolates were partially tested for the antibiotic panels defined herein. Human and environmental isolates were tested against up to 16 antimicrobials with the following relative concentrations (μg): ampicillin (AMP, 10), amoxicillin/clavulanic acid 2:1 (AMC, 20/10), azithromycin (AZI, 15), chloramphenicol (CHL, 30), cefotaxime (FOT, 5), cefoxitin (FOX, 30) ciprofloxacin (CIP, 5), ceftazidime (TAZ, 30), gentamicin (GEN, 10), nalidixic acid (NAL, 30), streptomycin (STR, 10), trimethoprim–sulfamethoxazole (SXT, 1.25/23.75), tetracycline (TET, 30), meropenem (MER, 10), tigecycline (TGC, 15), and sulfadiazine + sulfamerazine + sulfamethazine (SSS, 250). The antimicrobials’ panel used for wild boar isolates comprised up to 10 molecules and they were: ampicillin (AMP, 10), chloramphenicol (CHL, 30), ciprofloxacin (CIP, 5), ceftazidime (TAZ, 30), gentamicin (GEN, 10), meropenem (MER, 10), nalidixic acid (NAL, 30), trimethoprim–sulfamethoxazole (SXT, 1.25/23.75), tetracycline (TET, 30), and tigecycline (TGC, 15).

### 4.3. Statistical Analysis

All data were analysed using Stata 17 [72]. We applied the exact binomial test to calculate the prevalence of serotypes and the AMR and MDR of *Salmonella*, with 95% confidence intervals (CIs). Comparisons of serotypes’ diversity and AMR profiles between human and environmental isolates were generally evaluated with Pearson’s Chi squared test. In case of low numbers (e.g., the ‘Intermediate’ AMR category), Fisher’s exact test was the statistical test used for comparisons. For all statistical tests, a two-tailed significance level of α = 0.05 was adopted. 

## 5. Conclusions

This study demonstrated the circulation of a great diversity of *Salmonella* serotypes with variable AMR levels from different sources in Liguria, northern Italy. The diversity of *Salmonella* strains and AMR profiles found in surface waters is probably due to their contamination with bacteria from different sources, in addition to humans. Multidrug resistance patterns against high-priority critically important antibiotics suggests that the severity of this issue can affect human health. The resistance observed in environmental strains also marks the importance of the environment for the spread of antimicrobial resistance. Hence, the continuing monitoring of AMR should be performed in the long term to evaluate the spread of AMR in the natural environment. Moreover, the circulation of potential pathogenic *Salmonella* in wild boar populations confirms the possible role of this animal species in the transmission of the pathogen, while its role as amplifiers of MDR needs to be investigated in the future. 

## Figures and Tables

**Figure 1 pathogens-11-01446-f001:**
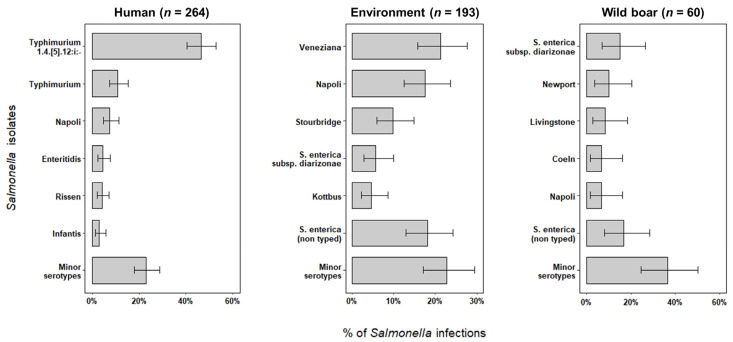
Prevalence and 95% CIs of most often recovered *Salmonella* isolates in human patients, environment and wild boars from Liguria, 2018–2020.

**Figure 2 pathogens-11-01446-f002:**
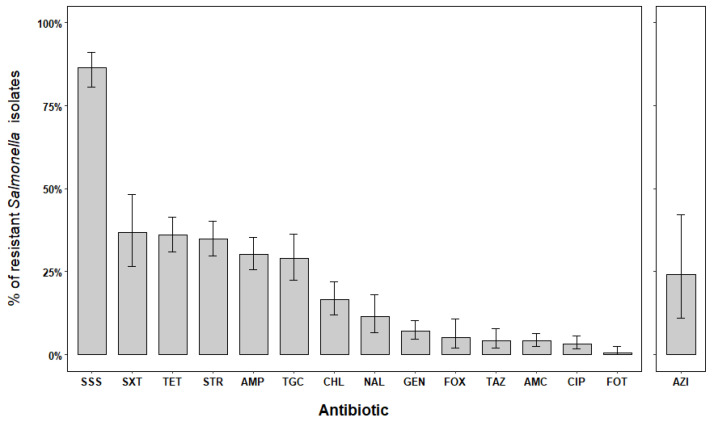
Overall percentage of AMR and 95% CIs in *Salmonella* isolates from Liguria, 2018–2020. Note: No resistant isolates were observed for meropenem (MER); resistance against azithromycin (AZI) was outnumbered and is thus represented independently from the rest of antibiotics. The names of the antibiotics were abbreviated as follows: AMC: amoxicillin/clavulanic acid, AMP: ampicillin, AZI: azithromycin, FOT: cefotaxime, FOX: cefoxitin, TAZ: ceftazidime, CIP: ciprofloxacin, CHL: chloramphenicol, GEN: gentamicin, NAL: nalidixic acid, STR: streptomycin, SSS: sulfadiazine + sulfamerazine + sulfamethazine, TET: tetracycline, TGC: tigecycline, SXT: trimethoprim–sulfamethoxazole.

**Table 1 pathogens-11-01446-t001:** Frequency and 95% CIs of *Salmonella* serotypes identified in samples collected from humans, wild boar and the environment, Liguria, 2018–2020.

Serotype/Subtype Identified	Human (*n* = 264)			Environment (*n* = 193)			Wild Boar (*n* = 60)		
	*n*	%	(95% CI)	*n*	%	(95% CI)	*n*	%	(95% CI)
*S. enterica* subsp. enterica (non-typed)	5	1.9	(0.6–4.4)	35	18.1	(13.0–24.3)	10	16.7	(8.3–28.5)
6,7:c:1,5	3	1.1	(0.2–3.3)	2	1.0	(0.1–3.7)	−	−	
*S.* Bovismorbificans	−	−		−	−		3	5.0	(1.0–13.9)
*S.* Brandenburg	5	1.9	(0.6–4.4)	1	0.5	(0.01–3.0)	−	−	
*S.* Bredeney	3	1.1	(0.2–3.3)	1	0.5	(0.01–3.0)	−	−	
*S.* Carno	1	0.4	(0.01–2.1)	−	−		−	−	
*S.* Cerro	−	−		1	0.5	(0.01–3.0)	−	−	
*S.* Choleraesuis var. Kunzendorf	−	−		−	−		3	5.0	(1.0–13.9)
*S.* Coeln	5	1.9	(0.6–4.4)	8	4.1	(1.8–8.0)	−	−	
*S.* Derby	3	1.1	(0.2–3.3)	−	−		−	−	
*S.* Drogana	1	0.4	(0.01–2.1)	−	−		−	−	
*S.* Eingedi	−	−		1	0.5	(0.01–3.0)	−	−	
***S.* Enteritidis**	**12**	**4.5**	**(2.4–7.8)**	−	−		**2**	**3.3**	**(0.4–11.5)**
*S.* Fann	−	−		1	0.5	(0.01–3.0)	−	−	
*S.* Give	3	1.1	(0.2–3.3)	−	−		−	−	
*S.* Goldcoast	3	1.1	(0.2–3.3)	1	0.5	(0.01–3.0)			
*S.* Hadar	1	0.4	(0.01–2.1)	−	−		−	−	
*S.* Ilugun	−	−		2	1.0	(0.1–3.7)	−	−	
***S.* Infantis**	**8**	**3.0**	**(1.3–5.9)**	−			−	−	
*S.* Kapemba	2	0.8	(0.1–2.7)	1	0.5	(0.01–3.0)	−	−	
*S.* Kentucky	1	0.4	(0.01–2.1)	−	−		−	−	
*S.* Kortrijk	1	0.4	(0.01–2.1)	−	−		−	−	
***S.* Kottbus**	−	−		**9**	**4.7**	**(2.2–8.7)**	**3**	**5.0**	**(1.0–13.9)**
*S.* Litchfield	1	0.4	(0.01–2.1)	−	−		−	−	
***S.* Livingstone**	1	0.4	(0.01–2.1)	−	−		**5**	**8.3**	**(2.8–18.4)**
*S.* London	3	1.1	(0.2–3.3)	1	0.5	(0.01–3.0)	−	−	
*S.* Mbandaka	1	0.4	(0.01–2.1)	−	−		−	−	
*S.* Muenster	1	0.4	(0.01–2.1)	−	−		3	5.0	(1.0–13.9)
***S.* Napoli**	**20**	**7.6**	**(4.7–11.5)**	**34**	**17.6**	**(12.5–23.7)**	4	6.7	(1.8–16.2)
***S.* Newport**	1	0.4	(0.01–2.1)	1	0.5	(0.01–3.0)	**6**	**10.0**	**(3.8–20.5)**
*S.* Othmarschen	2	0.8	(0.1–2.7)	−	−		−	−	
*S.* Panama	1	0.4	(0.01–2.1)	−	−		−	−	
*S.* Patience	1	0.4	(0.01–2.1)	−	−		−	−	
*S.* Presov	−	−		1	0.5	(0.01–3.0)	−	−	
*S.* Rissen	**11**	**4.2**	**(2.1–7.3)**	2	1.0	(0.1–3.7)	−	−	
*S.* Senftenberg	1	0.4	(0.01–2.1)	−	−		−	−	
S. Stoneferry	−	−		−	−		2	3.3	(0.4–11.5)
***S.* Stourbridge**	1	0.4	(0.01–2.1)	**19**	**9.8**	**(6.0–15.0)**	2	3.3	(0.4–11.5)
*S.* Thompson	3	1.1	(0.2–3.3)	8	4.1	(1.8–8.0)	1	1.7	(0.04–8.9)
*S.* Typhi	2	0.8	(0.1–2.7)	−	−		−		
***S.* Typhimurium**	**29**	**11.0**	**(7.5–15.4)**	1	0.5	(0.01–3.0)	−	−	
***S.* Typhimurium 1,4,[5],12:i:-**	**123**	**46.6**	**(40.5–52.8)**	5	2.6	(0.8–5.9)	1	1.7	(0.04–8.9)
*S.* Veneziana	1	0.4	(0.01–2.1)	**41**	**21.2**	**(15.7–27.7)**	−	−	
*S.* Virchow	1	0.4	(0.01–2.1)	−	−		−	−	
*S.* Wohlen	1	0.4	(0.01–2.1)	1	0.5	(0.01–3.0)	−	−	
***S. enterica* subsp. diarizonae**	1	0.4	(0.01–2.1)	**11**	**5.7**	**(2.9–10.0)**	**9**	**15.0**	**(7.1–26.6)**
*S. enterica* subsp. houtenae	−	−		1	0.5	(0.01–3.0)	2	3.3	(0.4–11.5)
*S. enterica* subsp. salamae	2	0.8	(0.1–2.7)	4	2.1	(0.6–5.2)	−	−	

Note: *Salmonella* serotypes in bold indicate the most frequent serotypes identified according to the origin.

**Table 2 pathogens-11-01446-t002:** Percentage of AMR in *Salmonella* isolates according to their origin and the antibiotic panel tested.

		No. of Isolates (%)							
SOURCE	*AMR*	AMC	AMP	AZI ^a^	FOT	FOX	TAZ	CIP	CHL
**HUMAN**	** *S* **	149/198 (75.3)	113/228 (49.6)	−	221/229 (96.5)	110/118 (93.2)	203/215 (94.4)	197/228 (86.4)	189/230 (82.2)
	** *I* **	34/198 (17.2) ***	4/228 (1.8)	23/23 (100) ***	7/229 (3.1)	2/118 (1.7)	3/203 (1.4)	21/228 (9.2) **	3/230 (1.3)
	** *R* **	15/198 (7.6) *	111/228 (48.7) ***	−	1/229 (0.4)	6/118 (5.1)	9/215 (4.2)	10/228 (4.4)	38/230 (16.5)
**ENVIRONMENT**	** *S* **	124/128 (96.8)	140/145 (96.5)	−	141/145 (97.2)	64/65 (98.5)	125/127 (98.4)	141/145 (97.2)	145/145 (100)
	** *I* **	2/128 (1.6) ***	3/145 (2.1)	2/10 (20.0) ***	4/141 (2.8)	1/65 (1.5)	2/127 (1.6)	2/145 (1.4) **	−
	** *R* **	2/128 (1.6) *	2/145 (1.4) ***	8/10 (80.0)	−	−	−	2/145 (1.4)	−
**WILD BOAR**	** *S* **	−	30/30 (100)	−	39/39 (100)	−	39/39 (100)	39/39 (100)	3/3 (100)
	** *R* **	−	−		−	−	−	−	−
		**No. of Isolates (%)**							
**SOURCE**	** *AMR* **	**GEN**	**MER ^a^**	**NAL**	**STR**	**SSS**	**TET**	**TGC ^b^**	**SXT ^b^**
**HUMAN**	** *S* **	173/230 (81.6)	32/33 (97.0)	100/139 (71.9)	84/208 (40.4)	5/117 (4.3)	81/206 (39.3)	47/99 (47.5)	7/21 (33.3)
	** *I* **	28/212 (13.2)	1/33 (3.0)	23/139 (16.5)	22/208 (10.6)	3/117 (2.6)	11/206 (5.3)	5/99 (5.1)	−
	** *R* **	11/212 (5.2)	−	16/139 (11.5)	102/208 (49.0) ***	109/117 (93.2) ***	114/206 (55.3) ***	47/99 (47.5) *	14/21 (66.7)
**ENVIRONMENT**	** *S* **	102/144 (70.8)	19/19 (100)	59/65 (90.8)	91/130 (70.0)	13/66 (19.7)	114/137 (83.2)	72/80 (90.0)	10/24 (41.7)
	** *I* **	28/144 (19.4)	−	6/65 (9.2)	23/130 (17.7)	4/66 (6.1)	13/137 (9.5)	3/80 (3.8)	−
	** *R* **	14/144 (9.7)	−	−	16/130 (12.3) ***	49/66 (74.2)	10/137 (7.3) ***	5/80 (6.2) *	14/24 (58.3)
**WILD BOAR**	** *S* **	39/39 (100)	35/35 (100)	4/4 (100)	−	−	4/4 (100)	35/35 (100)	36/39 (92.3)
	** *R* **	−	−	−	−	−	−		3/39 (7.7)

^a^ antibiotic only tested in 2020; ^b^ antibiotic tested in 2019 to 2020; ‘−‘ antibiotic not tested. The numbers marked with asterisks within the same column are significantly different for specific AMR levels *‘I’* (Fisher’s exact test **(*p* < 0.01); ***(*p* < 0.001)) and *‘R’* (* Pearson’s Chi squared test (*p* < 0.05); ***(*p* < 0.001)). Note: AMR levels were categorized into three main groups: susceptible (*S*), intermediate resistant (*I*), and resistant (*R*) strains. The name of antibiotics was abbreviated as follows: AMC: amoxicillin/clavulanic acid, AMP: ampicillin, AZI: azithromycin, FOT: cefotaxime, FOX: cefoxitin, TAZ: ceftazidime, CIP: ciprofloxacin, CHL: chloramphenicol, GEN: gentamicin, MER: meropenem, NAL: nalidixic acid, STR: streptomycin, SSS: sulfadiazine + sulfamerazine + sulfamethazine, TET: tetracycline, TGC: tigecycline, SXT: trimethoprim–sulfamethoxazole.

**Table 3 pathogens-11-01446-t003:** AMR profiles recorded among *Salmonella* strains resistant to quinolones and cephalosporines.

Serotype/Subspecies Involved	Source	No. of Strains	Cephalosporines and/or Quinolones	Antimicrobial Resistance Profile
6,7:c:1,5	human	1	Nalidixic acid	AMP-STR-SSS-**NAL**-SXT
S. Typhimurium 1,4,[5],12:i:-	human	1	Nalidixic acid	SSS-**NAL**
S. Typhimurium 1,4,[5],12:i:-	human	1	Ceftazidime	AMP-**TAZ**-STR-TET
S. Typhimurium 1,4,[5],12:i:-	human	1	Ceftazidime	AMP-**TAZ**-STR-SSS-TET
S. Typhimurium 1,4,[5],12:i:-	human	1	Cefoxitin Ceftazidime	AMP-**FOX**-**TAZ**-STR-SSS
S. Typhimurium 1,4,[5],12:i:-	human	1	Cefoxitin Ceftazidime	AMP-**FOX**-**TAZ**-STR-SSS-TET
S. Typhimurium 1,4,[5],12:i:-	human	1	Cefoxitin	AMC-AMP-**FOX**-CHL-STR-SSS-TET
*S*. Coeln	human	1	Ceftazidime	**TAZ**-SSS
*S. enterica* subsp. enterica	Environm.	1	Ciprofloxacin	AZI-**CIP**-GEN-SXT
*S*. Enteritidis	human	1	Nalidixic acid	SSS-**NAL**
*S*. Enteritidis	human	1	Nalidixic acid	AMP-STR-SSS-**NAL**
*S.* Goldcoast	Environm.	1	Ciprofloxacin	AZI-**CIP**-TET-TGC-SXT
*S.* Hadar	human	1	Nalidixic acid	AMP-STR-SSS-TET-**NAL**
*S.* Infantis	human	1	Ciprofloxacin	**CIP**
*S.* Infantis	human	3	Ciprofloxacin Nalidixic acid	**CIP**-STR-SSS-TET-**NAL**-SXT
*S.* Infantis	human	1	Nalidixic acid	SSS-TET-**NAL**
*S.* Infantis	human	1	Nalidixic acid	AMP-SSS-TET-**NAL**
*S.* Kentucky	human	2	Ciprofloxacin Nalidixic acid	AMP-**CIP**-GEN-STR-SSS-TET-**NAL**
*S*. Napoli	human	1	Ceftazidime	AMP-**TAZ**-CHL-STR-SSS-TET
*S.* Othmarschen	human	1	Cefoxitin Ceftazidime Ciprofloxacin	AMP-**FOX**-**TAZ**-**CIP**-SSS
*S.* Rissen	human	2	Ciprofloxacin Nalidixic acid	AMP-**CIP**-TET-**NAL**
*S.* Senftenberg	human	1	Cefotaxime	**FOT**
*S.* Thompson	human	1	Ceftazidime	**TAZ**
*S.* Typhi	human	1	Ciprofloxacin Nalidixic acid	**CIP**-**NAL**
*S.* Typhi	human	1	Ciprofloxacin Nalidixic acid	AMP-CHL-**CIP**-STR-SSS-**NAL**-SXT
*S.* Typhimurium	human	1	Cefoxitin	FOX-SSS
*S.* Typhimurium	human	2	Ceftazidime Ciprofloxacin Nalidixic acid	AMP-**TAZ**-**CIP**-CHL-STR-SSS-TET-**NAL**-
*S.* Typhimurium	human	1	Ciprofloxacin	AMC-AMP-**CIP**-CHL-GEN-STR-TET-TGC
*S.* Typhimurium	human	1	Nalidixic acid	AMC-AMP-STR-SSS-TET-**NAL**
*S.* Virchow	human	1	Nalidixic acid	SSS-**NAL**

Note: The names of antibiotics were abbreviated as follows: AMC: amoxicillin/clavulanic acid, AMP: ampicillin, AZI: azithromycin, FOT: cefotaxime, FOX: cefoxitin, TAZ: ceftazidime, CIP: ciprofloxacin, CHL: chloramphenicol, GEN: gentamicin, NAL: nalidixic acid, STR: streptomycin, SSS: sulfadiazine + sulfamerazine + sulfamethazine, TET: tetracycline, TGC: tigecycline, SXT: trimethoprim–sulfamethoxazole.

## Data Availability

All relevant data are provided in the present study or in Appendix A.

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
