# Peer review of "Antimicrobial Resistance of Salmonella Strains Isolated from Human, Wild Boar, and Environmental Samples in 2018–2020 in the Northwest of Italy"

_pathogens, 2022, doi:10.3390/pathogens11121446_

Round 1
Reviewer 1 Report
Results. Suggestions.
In general, the Results section needs a more exhaustive review. The way it is written detracts from the work. Remove redundant information of data presented in Resultas are shown in tables.
Rewrite the whole paragraph:
Out of 517 isolates collected, we excluded 22.3% (n= 102) of isolates that were ana- 123 lysed during 2020 following a different routine test method; the specimens concerned in- 124 cluded 47 Salmonella collected from the environment, 34 from human patients and 21 from 125 wild boar. Therefore, complete data on susceptibility testing of 415 isolates were used for 126 AMR evaluation. Susceptibility testing showed that nearly all isolates appeared susceptible to at least one antibiotic (n= 411).
Reviewer Note: The total number of samples tested must first be reported. The excluded percentaje must be at the end of the sentence.
Discussion.
Line 242: “veterinary matrices” what do you mean?
Line 243: Its presence… “the presence of S. Typhimurium” or The presence of Salmonella in swine”? Its not clear.
Line 279: mon. replace by monophasic
Could the observed difference in the diversity of serotypes in the environment be due to the low number of samples? A little discussion of this must be inlcuded.
Line 293 and 301: molecules? Use antibiotics.
Reviewer 2 Report
Listorti et al. investigated Salmonella recovered from humans, animals (wild boar), and the environment (water) for serotype diversity and AMR. One health in AMR is an important topic and the authors had good samples (from human, animals, and the environment) and results (serotypes and AMR rates and patterns) to make this manuscript a fine one. However, to increase the impact of the study, the authors could have done a better job of investigating the whole picture of AMR instead of just comparing AMR rates and patterns between sources and with those of other countries. Overall, the manuscript needs to be revised before it can be published.
I like that the authors were thorough, but the Result section seems too verbose.
- Try to be non-repetitive. For example, the results on the frequently recovered serotypes are presented as a figure (Fig 1), a table (Table 1 in bold), and the text. Please refrain from describing all the results if they are already presented as figures and tables. The text on resistance rates in page 5 can be greatly reduced since the information is presented in a table.
- Try to remove those that seem less important. For example, the information on the % resistance to each antibiotic agent that is shared by the isolates from human and animal, human and environment, etc can be deleted or greatly reduced (replace with a figure)
- For data on MDR (page 7), please reduce the text by presenting the data in a table or figure. The text is long and confusing.
A paragraph in the discussion section (lines 291-328) seems a bit long. The discussion section contains more grammatical errors than the other parts of the manuscript. Please revise thoroughly.
Here are specific comments:-
Line 56: delete “as a result of”
Line 58: resistance instead of resistances
Line 62: complete the sentence “AMR bacterial”
Line 60: the change of subject is a bit abrupt from talking about AMR transfer mechanism from bacteria to bacteria to the holistic picture of AMR in human, animal, and the environment. Consider adding a transitional word or sentence or separating the paragraphs.
Line 60: why was wild boar chosen as an indicator? Is wild boar prevalent in this region of the study?
Line 67: why did the authors mention different surveillance programs in this paragraph? Is this study a part of this program? If so, please mention here to emphasize the objective of the study.
Line 80: “related” or “corresponding” instead of “relative”?
Line 115: origin and sources are repetitive. Please revise the whole manuscript for the same error where the two terms “origin” and “source” are used together
Line 117: delete “and” after human and replace it with a comma
Line 131: again, antimicrobial resistance instead of antimicrobial resistances. Revise the whole manuscript for the same error i.e. “resistances”
Line 134: again, source of origin? Did you mean “source of the samples”
Line 154: All the abbreviations for antibiotics were given in parentheses but with commas here. Be consistent.
Line 155: what do you mean by “mildly resistant”? less number of isolates being resistant or MIC is lower?
Line 160: why not use abbreviation for meropenem? Be consistent
Line 168: spell out the number under 10. It should be “two types”
Line 173: the abbreviations should be at the footnote, not in the title
Table 2: What do S, I, and R indicate? Also, replace “N.” with “number” or “no.”
Table 2: why are the denominators for each antibiotic agent different? Were only a certain number of isolates tested for each agent? Please mention it in the method
line 204: resistance against to? Delete “to”
Table 3: the abbreviations on the table need to be capitalized to match the rest of the manuscript
Line 233: “as well as” instead of “but also”
Line 250: need to change “and is uncommonly” to either “but is uncommonly” or “and is commonly” for the sentence to make sense
Line 279: monophasic
Line 292: why not use the abbreviation AMR already introduced? Again, it should be “resistance” instead of “resistances”
Lines 301-303: there are two active verbs in this sentence. Please revise it
Lines 308-311: this sentence contains many grammatical errors e.g. it should be “In 2017” and “fluoroquinolones-resistant Salmonella”
Line 312: “… while for ciprofloxacin are, fortunately, lower”? Did you mean the AMR level against ciprofloxacin?
Lines 313-316: This sentence contains many grammatical errors, due to which the meaning of the sentence is unclear
line 321: “lower” what?
Line 330: delete “after the emergence of AMR against other antibiotics used for the salmonellosis treatment” which not only makes the sentence long and confusing but also unnecessary
Line 332 and 333: be more specific than “these molecules” and “this resistance”.
Line 336: what is “it” when the authors said “it needs to be carefully considered because…”
Line 343 and 345: again, what is “this molecule”? since it has been a while since you last mentioned what “this molecule” is, please specify. Also, throughout the manuscript, please be clear and specific, whenever possible
Lines 354-356: it should be “we also observed” rather than “also in our study we observed”. Also, it should be “we observed co-resistance to ciprofloxacin and ceftazidime in two strains of…”.
Line 379: 77.% of the strains
Line 384: promising for what?
Line 385: characterize
Line 433: what differences? Differences in diversity? AMR rates? AMR patterns?
Line 434: does the result of this study suggest that water contamination by “different pathogen sources” was the cause of the differences in Salmonella recovered from humans, animals, and the environment?
Line 436: fluoroquinolones
Line 437: spread instead of diffusion
Line 439: need a connecting word such as “and” before “in particular” because there are two complete sentences here
Line 389: please specify the “different sources” in the first line itself in the topic sentence e.g. “different sources including humans, animals, and the environment”
Line 392: specify what “our Institute” is. Also, use lower case for a common noun
Line 394: human-isolated strains
Line 396: Were Salmonella isolates from human and water samples given to the authors, but for the animal samples, did the authors processed the samples for Salmonella isolation? Could you please briefly mention that in the paragraph or specify in this sentence that the isolation was done in the present study?
Line 399: specify “the isolation of Salmonella”
Line 402: although the authors provided the reference, please briefly mention the serotyping method- was it PCR, antisera, sequencing, etc?
Line 404: how many wild boar samples? Also, since 264 Salmonella isolates from humans and 193 isolates from the environment do not add up to a total of 517 isolates, the authors should also specify the number of isolates recovered from the animals
Lines 412 and 418: “Thus” and “Instead” do not seem necessary. Please check the whole manuscript for the same error
Line 422: there are only 10 antibiotics while the authors mentioned that there were 12 of them
